# Validation of the Texas School Physical Activity and Nutrition (SPAN) Dietary Index Against the Healthy Eating Index Among Elementary-Aged Students

**DOI:** 10.3390/nu17121965

**Published:** 2025-06-10

**Authors:** Ethan T. Hunt, Allison N. Marshall, Raja Malkani, Nalini Ranjit, Adriana Pérez, David J. Badillo, Danielle J. Gartner, Ashley Schelfhout, Vijay R. Narayanan, Christopher D. Pfledderer, Deanna M. Hoelscher

**Affiliations:** 1Michael & Susan Dell Center for Healthy Living, University of Texas Health Science Center at Houston (UTHealth) School of Public Health in Austin, Austin, TX 78701, USA; 2School of Health, Gerogetown University, Washington, DC 20007, USA

**Keywords:** validation, Texas School Physical Activity and Nutrition (Texas SPAN), diet quality, Healthy Eating Index (HEI)

## Abstract

**Objective**: Assess the accuracy of the Texas School Physical Activity and Nutrition (SPAN) survey’s diet quality index against the 24 h recall-based Healthy Eating Index (HEI-2020). **Methods**: Fifty-one rising third and fourth graders (mean age 9.5 yrs., SD = 1.03 yrs.) from a summer program completed the SPAN survey and a 24 h dietary recall on the same day. The study compared SPAN HEI scores from survey food frequency items to HEI-2020 scores from recalls using Nutrition Data System for Research (NDS-R) software, evaluating correlations and agreement metrics. **Results**: SPAN HEI averaged 36.87 (SD = 3.78), while recall-derived HEI was 49.05 (SD = 11.92). The mean difference between indices was 12.18 (SD = 10.83), with an absolute difference of 13.51 (SD = 9.01). Bland–Altman analysis indicated limits of agreement from −9.05 to 33.40. Spearman correlation between SPAN HEI and recall HEI was r = 0.44 (*p* < 0.01), with an ICC of 0.45 (95% CI = 0.04, 0.68). **Conclusions and Implications**: After comparing HEI scores from both tools, SPAN HEI and HEI-2020 demonstrated a moderate correlation, indicating that SPAN HEI may serve as a practical and less burdensome alternative for large-scale dietary assessments. While further validation is needed, these findings suggest its potential utility in monitoring diet quality at the population level.

## 1. Introduction

Dietary patterns and behaviors are important contributors to overweight and obesity among children and adolescents [1,2]. Like the prevalence of obesity, dietary habits also track into adolescence and adulthood [3,4]. Because of this, accurate diet quality measurement is essential to appropriately monitor and examine dietary habits and trends among individuals and populations. The 24 h dietary recall is considered an acceptable criterion measurement method for dietary intake. However, in population studies, it is expensive, time-consuming, requires extensive training, and includes a significant burden and significant food knowledge to precisely recall diet during the previous 24 h [5].

The ongoing development of dietary intake tools must prioritize accuracy and reduce the data collection burden to effectively monitor diet quality at the population level. Simplified dietary assessment methods are essential for capturing contextual measures of diet, such as food or nutrition security, which ensures consistent and equitable access to healthy, safe, and affordable foods. By focusing on tools that are less time-consuming and burdensome, researchers can improve the feasibility of large-scale surveillance and intervention programs aimed at enhancing diet quality and addressing public health concerns related to food security, which have recently shown that households with children are at higher risk of food insecurity than the national average (12.5% compared to 10.2%), which is also associated with poor diet quality [6].

While a 24 h recall provides very detailed and accurate information, understanding overall diet quality at the population level can also provide important insight into eating habits, while reducing burden is an area to be explored in dietary research. Surveys and food frequency questionnaires (FFQs) are less burdensome to participants, are more feasible for larger sample sizes, and can be used to construct diet quality indices or represent dietary patterns that promote better health outcomes. Diet quality indices are useful tools for understanding overall diet quality and how dietary patterns compare to recommendations [7]. A variety of diet quality indices exist, including the HEI-2020, which is designed to assess diet quality relative to the United States Department of Agriculture Dietary Guidelines for Americans [8,9]. Short dietary assessment tools that provide a ‘snapshot’ of total diet quality similar to the HEI-2020 would be helpful for surveillance studies, which are often time–cost-limited but can be useful in tracking trends over time [10].

The 2022–2023 Texas School Physical Activity and Nutrition (Texas SPAN) instrument, designed to assess nutrition behaviors among 4th-, 8th-, and 11th-grade students, has been previously validated for its accuracy in measuring the intake frequencies of various food items [11,12,13]. This validation involved comparing SPAN survey responses with those from 24 h dietary recalls and other established dietary assessment tools, demonstrating the reliability and validity of the SPAN questionnaire for dietary assessment in school settings [11,13,14]. The SPAN survey uses short-term recall to measure the consumption of various foods consumed the previous day. This type of short-term recall is better for children because of the level of cognitive skill development needed to estimate averaging and frequency compared to a traditional food frequency survey [15]. Previous versions of the SPAN 4th-grade questionnaire capturing nutrition outcomes have been tested for reliability. Results indicated that it is reliable for food choice behaviors but less reliable for nutrition knowledge and attitudes [11,12,13,14]. To provide a more holistic view of diet quality, the SPAN survey has been used to develop a SPAN Healthy Eating Index (SPAN HEI) in multiple studies [16,17,18].

Despite good content validity in several studies [12,16,17], the SPAN HEI has yet to be validated against a criterion measure of diet quality, such as the HEI-2020. The primary objective of this study was to validate a diet quality measure derived using the most current version of the SPAN dietary intake survey among rising third- and fourth-grade students attending a summer camp.

## 2. Materials and Methods

SPAN was developed as a surveillance instrument to measure physical activity, nutrition behaviors, and attitudes in children and adolescents [13]. Two questionnaires were developed to measure these constructs at different developmental levels of students: one for 4th-grade students and a second for 8th- and 11th-grade students [11,13,14]. For this project, we measured the dietary intake of rising 3rd- and 4th-grade students because the summer program where the study took place exclusively served elementary-aged children, which included rising 3rd and 4th graders but did not include 8th- or 11th-grade students. The 4th-grade survey instrument included pictures to aid student comprehension and was determined to have a reading level appropriate for a nine-year-old child based on the Flesch–Kincaid readability test, cognitive interviews, and focus groups [19].

During the summer of 2022, one elementary school in Texas agreed to participate in this study over four weeks. This opt-in convenience sampling strategy was based on a non-probability strategy where only students enrolled in the summer program were invited to participate in the study. Students were included in the study if they returned a signed parental consent form to study staff during the study period. The protocol and survey instrument were fully approved by the first author’s institution’s Committee for the Protection of Human Subjects and administrators from the participating school and summer program. Because this project was under the overall ongoing Texas SPAN, which had undergone full institutional review, this protocol underwent an expedited review. The study was conducted in accordance with the Declaration of Helsinki and the ethical principles outlined in the Publication Manual of the American Psychological Association, and was approved by the Institutional Review Board of The University of Texas Health Science Center at Houston (protocol code HSC-SPH-21-0212; approved May 2021). Written consent from the parents/guardians and verbal assent from the students were obtained before collecting data. Before data collection, research personnel attended the first week of the 2022 summer program to actively invite parents and students to participate by distributing approved research flyers and consent forms to parents/guardians at pick-up/drop-off. At this time, parents/guardians were allowed to ask any additional questions regarding the study and its protocols. This study utilized a convenience sampling approach based on participation in a structured summer camp program. No a priori sample size calculation or statistical power analysis was performed. Power limitations are acknowledged in the Discussion section.

The SPAN questionnaire is intended to measure the food consumed over the previous day; therefore, a single 24 h recall was selected as the criterion measure administered to participants on Tuesday through Friday. An estimated total of 12 classrooms were occupied throughout the summer program. The specific day was coordinated with the summer program, as the SPAN instrument is meant to capture dietary intake on weekdays at school. This data collection structure was implemented because the SPAN survey captures school-related behaviors and dietary habits, which tend to vary between weekdays and weekends [13]. To be consistent, data were not collected on the day following a holiday [11,14,16,18].

Demographic characteristics (e.g., age, sex, and date of birth) were measured with five questions completed by the students. The 2022–2023 4th-grade SPAN questionnaire measures 26 individual food and dietary habit-related questions using a non-quantified food frequency format [5]. Food behaviors measured included intake of high-fat food items, high-calorie/low-nutrient food items, dairy products, fruits, vegetables, meat/fish/poultry, and grain products consumed during the previous day. Response options captured the number of times each item was consumed the previous day: 0 times, 1 time, 2 times, or 3 or more times.

Of the 26 food items, 13 were classified as “healthy” and included baked chicken/fish, nuts, brown rice, brown/wheat bread (e.g., whole grain bread), starchy vegetables (e.g., baked potatoes, but not French fries or chips), orange vegetables/carrots, green vegetables/salad, other vegetables, beans/legumes, fruit (fresh or canned), plain milk, yogurt, and 100% fruit juice. The remaining 13 food items were classified as “unhealthy” and included red meats, fried meats, white rice, white bread, French fries, frozen desserts, baked goods/sweet rolls, candy, flavored milk, fruit punch or fruit drinks, soda, sweetened coffee, and energy drinks.

All research assistants involved in the project were previously trained for SPAN survey administration, including survey protocols described elsewhere [13,20,21]. Prior to data collection, staff was also trained on the assessment of the 24 h recall during a two-day training session from the Nutrition Data System for Research (NDS-R) through the Nutrition Coordinating Center at the University of Minnesota, Minneapolis. Before data collection, interviewers practiced the techniques for two weeks with twenty volunteers, including children and adults, in monitored interviews. Variability in data collection was discussed, and consensus on interpretation was reached between staff members who would be collecting project data. During data collection interviews, notes were made by the interviewer on any unclear items; these items were discussed among the trained interviewers at a later time to arrive at a consensus for consistent interpretation. As with previous validations, the purpose of training and practicing was to ensure standardization during the actual interviews for the 24 h recall interview [13,20].

The 24 h recalls were collected during face-to-face interviews using standard NDS-R software version 2022 protocols. This protocol used a multiple-pass approach to reduce underreporting and to aid in further standardization of the interviews [22,23]. Food items were directly entered into the computer by name or using the NDS-R paper materials and then transcribed into the computer software by trained research assistants on the same day of data collection. The research staff assisted the children in completing the SPAN questionnaire in a quiet classroom. Research staff instructed participants to read the SPAN survey to themselves and ask for help with reading comprehension using a standard protocol that included examples of how to complete the questionnaire.

All participating students were randomly divided into two groups to complete the 24 h recalls and surveys. Participants started their study day with either the SPAN survey or the 24 h recall. They completed the remaining measure in the afternoon, with a period of two or more hours between measurements. Maintaining at least a two-hour difference between recall and survey administration was necessary to reduce the memory effects of the previous measurement [24]. Between the administration of the two instruments, participants adhered to their regular schedules and ate lunch, which is likely to have further reduced memory effects [25]. Eight to sixteen students participated in measurements each day. Participants took approximately 15–20 min to complete the SPAN survey and 25–35 min to complete the dietary recall interview. Water bottles and gym bags were distributed as incentives after completing their final measurement. Further, at the end of each measurement day, research assistants also distributed gift cards to parents/guardians as incentives for their children’s participation in the study.

For quality control, the 24 h recall and SPAN survey data were reviewed at the end of each day. In addition, the 24 h recall data were also checked for plausibility with established protocols to exclude subjects with energy intake of less than 500 and exceeding 5000 kcals; none of the participants in this study was excluded from either the SPAN survey or the 24 h recall [26].

All statistical analyses were performed using Stata software version 16 (StataCorp, College Station, TX, USA). We described descriptive statistics based on percentages for categorical outcomes and means and standard deviations for continuous variables. Following previous protocols, several steps were followed to organize and analyze data. SPAN questionnaires were examined for multiple markings or other evidence of discrepancies that could invalidate them for the purpose of the study [11,13]. The SPAN HEI was computed by calculating the number of times each subject reported consuming various food types on the previous day, where each answer choice ranged from 0 times (scored as 0) to 1 time (1), 2 times (2), or 3 or more times (3). Each food item has a possible score of 0–3. Healthy food items were summed and unhealthy foods were reverse-coded and summed before computing the total composite score for each subject. The SPAN HEI was then computed by scaling the total score such that the maximum possible score was 100 [16].

The 24 h dietary recall reports were reviewed and checked for errors after being completed after each collection day. Research assistants obtained a menu from the school food service personnel daily to cross-check responses as needed. The 24 h dietary recall data from the NDS-R software food and meal output files were obtained for each student and grouped into food categories, and an NDS-R-derived HEI-2020 was calculated for each child. The HEI-2020 scores from the NDS-R were calculated based on detailed dietary intake data collected through 24 h dietary recalls, including multiple components representing adequacy and moderation of dietary intake. This differs from the SPAN HEI, which sums the frequencies of healthy food items and reverse-codes unhealthy items to produce a composite score [8,9].

Data were analyzed using the Spearman rank-order correlation coefficient, and the percentage agreement was estimated along with the Bland–Altman plots [27,28]. Spearman rank-order correlations were computed because data were not normally distributed [29]. Bland–Altman plots were used as a graphical representation that allows for the assessment of agreement between the SPAN HEI and 24 h recall Hel-2020 [27,28,30]. An intraclass correlation coefficient (ICC) is presented to quantify the degree of consistency between the two diet quality measures measuring the total variability in the measurements [31]. The ratio used to calculate the ICC was the variance of interest (i.e., between-subject variance) divided by the total variance, including between-subject and residual variance. Since our data came from twelve classrooms within the school, we utilized a mixed modeling approach with ICC calculated based on variance components to accurately reflect this structure. The ICC calculation was performed using Stata’s *icc* command. This command utilizes a two-way random effects model to assess consistency across measurements, which indicates no reliability beyond what would be expected by chance [32]. Finally, both the difference and absolute difference between measures are presented to show the magnitude and direction of agreement and provide a basis for practical implication. The ‘difference’ between the SPAN HEI and HEI-2020 scores was calculated by subtracting each participant’s SPAN HEI score from the HEI-2020 score. The ‘absolute difference’ was calculated using the absolute value of this difference, providing a measure of the magnitude of the discrepancy between the two scores, regardless of whether the SPAN HEI score was higher or lower than the HEI-2020 score. While both metrics offer insights into agreement, they serve distinct purposes: mean difference highlights systematic bias, whereas mean absolute difference evaluates overall deviation. The consistency between the SPAN HEI and 24 h recall HEl-2020 was estimated for the overall sample, as well as stratified by sex [33,34]. In this study, we refer to validity as the degree to which the SPAN-HEI score approximates the HEI-2020 score, a commonly used reference method. Reliability refers to the consistency of SPAN-HEI scores across repeated applications. The terms accuracy and stability are used more broadly to describe the alignment of SPAN-HEI scores with expected dietary quality measures and their reproducibility.

## 3. Results

Descriptive and demographic information on the final sample can be found in Table 1. The final sample included 51 participants (59% girls, mean age 9.50, SD = 1.03 years). A majority of the participants in the sample (96%) self-identified as non-White Hispanic. The overall average SPAN HEI score was 36.87 (SD = 3.78). When stratified by sex, SPAN produced mean SPAN HEI scores of 37.96 (SD = 3.62) and 35.32 (SD = 3.51) for girls and boys, respectively. The 24 h recall produced a total sample mean HEI-2020 score of 49.05 (SD = 11.92). When stratified by sex, the 24 h recall produced mean HEI-2020 scores of 50.77 (SD = 9.65) and 46.60 (SD = 14.47) for girls and boys, respectively. The average difference and absolute difference between the two measures (24 h HEI-2020 and SPAN HEI) produced a mean difference of 12.18 (SD = 10.83) and an absolute difference of 13.51 (SD = 9.01), respectively.

### 3.1. Spearman Correlations Coefficients and Intraclass Correlation Coefficients

The Spearman correlation coefficients can be found in Table 2. The overall correlation between the SPAN HEI and the 24 h recall HEI-2020 was *r* = 0.44, *p* < 0.01; for girls it was *r* = 0.58, *p* < 0.01 and for boys it was *r* = 0.22, *p* = 0.35. The intraclass correlation coefficients (ICCs) can also be found in Table 2, with results similar to the Spearman correlation coefficients for the overall sample and when compared between girls and boys.

### 3.2. Percentage Agreement

Figure 1 illustrates the Bland–Altman analysis by sex, with the mean of both SPAN HEI and 24 h recall HEI-2020 scattered on the x-axis and the difference between the SPAN HEI and 24 h recall HEI-2020 scattered on the y-axis. Mean bias and limits of agreement (LOA) are also plotted. Mean bias is the average difference between the SPAN HEI and HEI-2020 scores, reflecting the direction and magnitude of systematic differences between the two measurement methods. Limits of agreement were calculated using a 95% confidence level of the differences between the two methods to understand the level of agreement and consistency between the SPAN HEI and HEI-2020 scores [28]. The mean bias of the SPAN HEI underestimated 24 h recall HEI-2020 by 12.18 points, representing a 25% relative difference between 24 h recall and SPAN HEI, with lower and upper levels of agreement ranging from −9.05 to 33.4. In addition to the mean differences and LOAs produced, the Bland–Altman plot also depicts that all individuals who fell outside the upper and lower LOAs were boys. This highlights the lack of agreement between the SPAN HEI and 24 h HEI-2020 measures in boys.

## 4. Discussion

In this study, our goal was to validate a measure of diet quality using the SPAN HEI compared to the HEI-2020 calculated from the Texas SPAN dietary survey in a convenience sample of elementary-aged students attending a summer program. Our findings indicate that the SPAN HEI produces a measure of diet quality with similar findings compared to the 24 h recall-derived HEI-2020. Comparisons between the SPAN HEI and the HEI-2020 showed correlations above 0.4 only among elementary school girls, indicating that the relationship between these diet quality measures may vary by sex. These findings suggest the potential for sex differences in dietary reporting, which should be further investigated in future studies. Our findings align with previous research that has observed variations in dietary reporting accuracy and diet quality perceptions between boys and girls [35,36]. Although the strongest alignment between the SPAN HEI and HEI-2020 was observed among girls, the overall findings provide important preliminary evidence supporting the potential utility of the SPAN HEI for dietary surveillance in young people. Given the increasing demand for low-burden, scalable tools to monitor dietary behaviors, particularly in school and community settings, this work contributes to the growing evidence base. Further validation in more diverse and representative samples, using multiple 24 h recalls, is warranted. Our findings indicated that the SPAN HEI scores differ from HEI-2020 scores by approximately 12.2 points in elementary-aged children. The lower score could be attributed to the smaller and more targeted number of foods assessed in the SPAN instrument compared to the unlimited and more precise recording of foods as part of the 24 h recall. The Spearman correlation coefficients indicate a moderate correlation for the overall sample and for girls, whereas no correlation was found for boys between SPAN HEI and the 24 h recall HEI-2020. Results from girls tended to be more consistent than those from boys. In addition, the sample included a greater percentage of girls than boys (58.9% versus 41.1%). This finding is in agreement with the literature that has highlighted sex differences concerning nutrition knowledge and efficacy [33,34].

The overall correlations and those for girls were within the range of other validated pediatric dietary intake questionnaires and surveys [37,38,39]. Our study found an overall ICC of 0.45, suggesting a level of consistency over time that falls on the upper end of what has been classified as ’poor’ reliability [40]. While not indicative of strong reliability, this aligns with expectations for dietary recall-based measures in this setting, which inherently capture variability in eating patterns over time [12]. Future research should explore strategies to enhance measurement stability, such as refining dietary assessment tools or incorporating repeated measures to better account for within-person fluctuations. Bland–Altman analysis showed moderate bias and limits of agreement. Within the Bland–Altman figure, it appears that sex was a potential driver in the most considerable differences depicted. Only boys in our Bland–Altman plot fell outside the limits of agreement on both the lower and upper end. This, paired with differences in correlation by sex, highlights dietary intake differences by sex that have been found in previous studies [34,37]. SPAN questionnaires are usually administered to 4th-grade students. Since this study was conducted over the summer, some of the students in this study were younger than the usual population. The SPAN survey has also been administered in 8th- and 11th-grade populations; replicating the validation study in older age groups might result in better correlations.

The USDA has prioritized food and nutrition security through food assistance programs such as the Supplemental Nutrition Assistance Program and the National School Lunch and Breakfast programs [41,42,43,44]. These programs will need measurement tools that capture diet quality among children and adolescents. Because of its limited burden, cost, and time to complete, the SPAN HEI could be a potential option when working to capture population-level estimates of food security and diet quality in elementary school students, especially in girls. In particular, the SPAN HEI scores show similar dietary quality rankings to the HEI-2020, which may be useful in characterizing group-level differences. In addition, other validated measures are also available and warrant discussion. The ASA24 (Automated Self-Administered 24 h Recall) is an online tool developed by the National Cancer Institute to collect detailed dietary intake data through a 24 h recall. While the ASA24 is a free, validated tool for dietary assessment in children, it requires participant training and significant staff support, making it less feasible for elementary school children in a school setting. In contrast, the SPAN HEI, with its FFQ format, is designed explicitly for self-administration with minimal training and is particularly useful in environments with limited access to technology or internet connectivity, making it a more practical option for large-scale studies [45,46]. Further research scoring the SPAN HEI with different weight scores may generate higher reliability with the 24 h recall. The use of ’healthy’ and ’unhealthy’ food items in short FFQs serves different purposes depending on the context. The SPAN HEI represents a dietary pattern that includes higher intakes of baked chicken/fish, nuts, brown rice, brown/wheat bread (e.g., whole grain bread), starchy vegetables (e.g., baked potatoes, but not French fries or chips), orange vegetables/carrots, green vegetables/salad, other vegetables, beans/legumes, fruit (fresh or canned), plain milk, yogurt, and 100% fruit juice, and lower intakes of foods higher in sugar, saturated fat, and sodium, including red meats, fried meats, white rice, white bread, French fries, frozen desserts, baked goods/sweet rolls, candy, flavored milk, fruit punch or fruit drinks, soda, sweetened coffee, and energy drinks. The foods in the SPAN dietary pattern (e.g., the SPAN HEI) are consistent with those identified in the Dietary Guidelines for Americans, which have been found to promote better health outcomes consistently. The SPAN HEI provides a measure of dietary quality that may contribute to an understanding of overall food and nutrition security in broader assessments, though further validation is needed. Using FFQs that capture both healthy and unhealthy food consumption is essential for addressing optimal food security and preventing diet-related chronic diseases [47,48,49,50].

This study has several strengths that should be noted. First, we included multiple statistical methods to provide a foundation for results that have been used in previous dietary intake validation studies [51,52]. Second, estimating the correlation between the criterion standard of the 24 h recall and the recall scores using the Spearman rank correlation coefficient does not require data to be distributed normally. Additionally, incorporating percentage agreement via Bland–Altman analysis illustrates mean bias, limits of agreement, and trend direction of the data [29,30]. Finally, we incorporated an established standard criterion comparison method using NDS-R-derived HEI-2020. Using the 24 h recall, we allowed for the most precise comparison to our SPAN survey, which has also been incorporated in other studies examining infants and young children [53].

However, this study has several limitations. The sample size is relatively small and consists of a homogeneous group of students, including rising 3rd and 4th graders, during a small summer session at one school. As a result, the findings may not be generalizable to broader elementary school populations. Participation in the study was voluntary and required both parent/guardian consent and student assent, which presented logistical barriers that likely contributed to the low enrollment rate. This limited participation should be interpreted as a constraint on the generalizability and scale of the findings and underscores the challenge of recruiting young people in school or community-based settings. As such, the findings should be interpreted with caution, as they may reflect limited statistical power rather than a true absence of effect. To aid interpretation despite these limitations, Bland–Altman plots were included to provide a visual representation of agreement and variability between dietary measures. In addition, only a single 24 h dietary recall was collected per participant, which may be sufficient for estimating average intake in large populations but is suboptimal for assessing individual dietary intake or for analyses involving small subgroups. This limitation may introduce day-to-day variability that is not accounted for in our comparison. Another potential limitation of this study is the voluntary nature of participation in the summer program, which may affect the representativeness of the sample. Families who choose to enroll their children in such programs may differ systematically from those who do not, potentially influencing dietary behaviors and attitudes. Factors such as greater parental involvement or specific educational goals could contribute to differences in dietary intake, limiting the generalizability of the findings to all elementary school children. Additionally, among boys, the correlations were lower than 0.4, which may be partially attributed to the limited sample size. The SPAN survey applies a uniform frequency scale (0, 1, 2, or 3+ times per day) to all foods, regardless of dietary recommendations. While designed for simplicity in large-scale surveillance, this approach does not account for differences in optimal intake levels across food groups. As a result, SPAN HEI scores may differ from HEI-2020, where specific foods contribute variably to diet quality. Future research should explore alternative scoring methods that better align with dietary guidelines. Future research should also consider using larger sample sizes to improve the precision of the estimates and explore potential differences by sex. Further, we acknowledge that the SPAN HEI scoring methodology, which equally weighs healthy and unhealthy items, does not fully align with the more nuanced sufficiency and moderation domains of the HEI-2020. This difference, and the lack of adjustment for energy intake in the SPAN HEI, may limit the comparability of the two measures and should be considered when interpreting the results. Further research scoring the SPAN HEI with different weight scores may generate higher reliability with the 24 hr recall. We also did not formally employ an ICC to assess inter-rater reliability. However, to ensure consistency, all researchers underwent two weeks of intensive training in NDS-R techniques, with 20 practice recall sessions with children and adults. This training involved consensus checking to resolve discrepancies and in-depth note-taking to maintain accuracy and standardization. Finally, both the SPAN survey and the 24 h recall require accurate recognition of food items and accurate memory of food items eaten the previous day, which presents limitations, especially in this age group. The reliance on memory for dietary recall can introduce recall bias, where participants may forget or misreport foods consumed. Studies have shown that recall bias can vary by sex, with girls often providing more accurate dietary recalls than boys, possibly due to higher levels of attention to dietary habits or greater interest in health and nutrition [5]. This could explain why girls’ reports were more closely aligned with the SPAN HEI and HEI-2020 measures than boys’. The alignment in girls’ reports suggests they may have a higher level of engagement or accuracy in recalling dietary intake. In contrast, boys may exhibit more variability, leading to discrepancies between the two tools. Coupling these recalls and interviews with direct observation may increase our ability to accurately assess intake. However, direct observation also presents challenges, including the need for significant time and resources, such as trained personnel, the risk of observer bias, and the potential for participants to alter their behavior due to being observed, a phenomenon known as the Hawthorne effect [54]. Additionally, direct observation can be intrusive and is not always feasible in all settings, particularly in large-scale studies [55,56]. Given the limitations surrounding recall, researchers have proposed using biomarkers to provide a complimentary objective measure of capturing information on dietary intake, but assessing food intake using specific biomarkers may present a challenge in recruiting children and families [57], and it can be unclear exactly which biomarkers to use to represent total dietary intake.

## 5. Conclusions

Our analysis was designed to provide group-level validity estimates of a measure of diet quality using the SPAN HEI, derived from the SPAN survey instrument. These findings suggest that the SPAN-HEI, while limited by its structure, may serve as a practical alternative for monitoring dietary quality in large population-based youth studies, particularly where 24 h dietary recalls are not feasible due to time or resource constraints. However, its applicability to other elementary-grade populations, especially boys, requires further investigation. Its strengths of relatively low time to complete, high ease of use, and comprehensive diet assessment make it a tool for consideration in field use when collecting group-level estimates of child dietary intake and subsequent nutrition quality. While 24 h dietary recalls are considered an acceptable criterion for dietary assessment, they have several limitations that make them impractical in specific settings. For large-scale population studies, 24 h recalls are often time-consuming, require significant training for accurate administration, and can be costly [58]. Given the difficulty and expense of administering 24 h recall interviews in large samples, these data suggest the SPAN dietary intake survey may serve as a complementary tool for assessing diet quality at the group level among 4th-grade students, especially when examining relative dietary quality and patterns among populations or over time. Further evaluation should be performed to refine the instrument and index further, replicating these results in middle and high school students, as well as more diverse populations.

## Figures and Tables

**Figure 1 nutrients-17-01965-f001:**
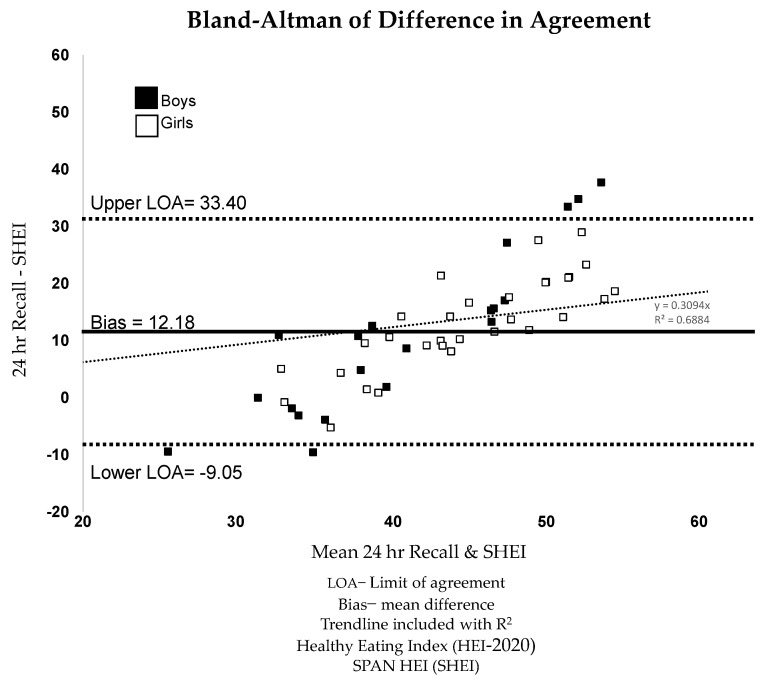
Bland–Altman difference in agreement examining SPAN survey HEI and 24 h. recall HEI.

**Table 1 nutrients-17-01965-t001:** Descriptive and demographic information on validation sample (N = 51).

Girls (n, %)	30	58.90%
**Age (mean, ±),**		
Overall (N = 51)	9.50	1.03
Girls (n = 30)	9.50	1.10
Boys (n = 21)	9.45	0.94
**SPAN HEI** **Scores (0–100)**	Mean	SD
Overall	36.87	3.78
Girls	37.96	3.62
Boys	35.32	3.51
**24 Hr. Recall HEI-2020 scores (0–100)**		
Overall	49.05	11.92
Girls	50.77	9.65
Boys	46.60	14.47
**24 Hr. Recall HEI-2020–SPAN HEI (difference)**	12.18	10.83
**24 Hr. Recall HEI-2020–SPAN HEI (absolute difference)**	13.51	9.01
Healthy Eating Index (HEI-2020) School Physical Activity and Nutrition Healthy Eating Index (SPAN HEI)		

**Table 2 nutrients-17-01965-t002:** Spearman rank-order and intraclass correlation coefficients examining SPAN HEI and the 24-h recall Healthy Eating Index (HEI-2020).

Spearman Rank-Order Correlation	Coef.	*p*-Value			
Overall	0.44	<0.01 *			
Girls	0.58	<0.01 *			
Boys	0.22	0.35			
**Intraclass Correlation Coefficient**	**Coef.**	**SE.**	**95% CI**		***p*-value**
Overall	0.45	0.12	0.04	0.68	0.02 *
Girls	0.62	0.18	0.21	0.82	0.01 *
Boys	0.25	0.17	−0.81	0.69	0.26

* indicates a *p*-value < 0.5 and a *p*-value of <Type I errors of 0.05 when testing the null hypothesis that there is no correlation (i.e., the two variables are independent). Healthy Eating Index (HEI-2020).

## Data Availability

The original contributions presented in this study are included in the article. Further inquiries can be directed to the corresponding author.

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
