# Peer review of "Validation of the Texas School Physical Activity and Nutrition (SPAN) Dietary Index Against the Healthy Eating Index Among Elementary-Aged Students"

_nutrients, 2025, doi:10.3390/nu17121965_

Round 1
Reviewer 1 Report
Comments and Suggestions for Authors
This is a very good and well-organized article. However, I would like to raise a few points for clarification and improvement:
1., the authors mention the Declaration of Helsinki, but it is not clear whether the study also followed the ethical standards described in the Publication Manual of the American Psychological Association. If the study did follow those APA ethical principles, please state this clearly in the article.
2., while the conclusions are well written, I recommend making the practical implications of the study more visible and direct.
3., the references should be updated to follow the MDPI Style and Guidelines. Please make sure all in-text citations and the reference list are correctly formatted.
Author Response
We thank the editor and the peer reviewers for their time and expertise in the review of our submission. We have reviewed and responded to each comment provided by both peer reviewers below. Responses to comments are provided and bolded. The resubmitted documents and manuscripts (both clean and with track changes) provides all additional information and addressed feedback from reviewers using track changes.
Reviewers' comments to author:
Reviewer 1.
This is a very good and well-organized article. However, I would like to raise a few points for clarification and improvement:
1., the authors mention the Declaration of Helsinki, but it is not clear whether the study also followed the ethical standards described in the Publication Manual of the American Psychological Association. If the study did follow those APA ethical principles, please state this clearly in the article.
We have revised the Institutional Review Board Statement to clarify that the study followed both the ethical standards outlined in the Declaration of Helsinki and the APA ethical principles. The revised sentence now reads:
The study was conducted in accordance with the Declaration of Helsinki and the ethical principles outlined in the Publication Manual of the American Psychological Association, and was approved by the Institutional Review Board of The University of Texas Health Science Center at Houston (protocol code HSC-SPH-21-0212; approved May 2021).
2., while the conclusions are well written, I recommend making the practical implications of the study more visible and direct.
To address this, we have added the following in the conclusion:
“These findings suggest that the SPAN-HEI, while limited by its structure, may serve as a practical alternative for monitoring dietary quality in large population-based youth studies, particularly where 24-hour dietary recalls are not feasible due to time or resource constraints.”
3., the references should be updated to follow the MDPI Style and Guidelines. Please make sure all in-text citations and the reference list are correctly formatted.
It appears that the version of the manuscript that was formatted and sent back from Nutrients has stripped the referencing software, disabling our ability to update our references to meet the MDPI guidelines. Editors, please advise.
Reviewer 2 Report
Comments and Suggestions for Authors
The manuscript is well presented and the methods adequately described. However, we find a few issues that need addressing:
- The concepts of accuracy, validity, reliability, consistency and stability of measurements are used in several places and would need to be clearly defined.
- The 24h food recall (or the HEI-2000) are NOT gold standard methods to assess dietary intake.
- What about sample size determination or statistical power?
- Food security should be used instead of nutrition security.
- Many factors impinge on the representativeness of the findings: small sample size, summer camp context, a majority of hispanic participants, but also only one 24h food recall, which may be sufficient for large population groups but not for small groups or individuals, whcih would require three recalls.
- The short method appears valid only for girls until further validation work is done on a larger and more representative sample, using as reference 3 24h recalls instead of one. So what is the practical relevance and importance of this paper?
Author Response
We thank the editor and the peer reviewers for their time and expertise in the review of our submission. We have reviewed and responded to each comment provided by both peer reviewers below. Responses to comments are provided and bolded. The resubmitted documents and manuscripts (both clean and with track changes) provides all additional information and addressed feedback from reviewers using track changes.
Reviewers' comments to author:
Reviewer 2.
The manuscript is well presented and the methods adequately described. However, we find a few issues that need addressing:
The concepts of accuracy, validity, reliability, consistency and stability of measurements are used in several places and would need to be clearly defined.
We have clarified the use of these terms in the revised manuscript. Specifically, in the Methods section, we have added the following:
In this study, we refer to validity as the degree to which the SPAN-HEI score approximates the HEI-2020 score, a commonly used reference method. Reliability refers to the consistency of SPAN-HEI scores across repeated applications. The terms accuracy and stability are used more broadly to describe the alignment and reproducibility of SPAN-HEI scores with expected dietary quality measures.
The 24h food recall (or the HEI-2000) are NOT gold standard methods to assess dietary intake.
Throughout the manuscript, we edited language to state how the recall is an acceptable criterion over a gold standard.
What about sample size determination or statistical power?
This study used a convenience sample drawn from a summer program cohort, and as such, no formal sample size calculation or power analysis was conducted. We have added this clarification to the Methods and explicitly acknowledge statistical power as a study limitation in the Discussion section.
As such, findings should be interpreted with caution, as they may reflect limited statistical power rather than a true absence of effect. To aid interpretation despite these limitations, Bland-Altman plots were included to provide a visual representation of agreement and variability between dietary measures.
Food security should be used instead of nutrition security.
We appreciate the reviewer’s suggestion and the opportunity to respectfully clarify this distinction. We intentionally used the term “nutrition security” rather than “food security” to reflect the evolving public health focus on not just food access, but access to nutritionally adequate, health-promoting food. This terminology is aligned with the USDA and other federal agencies, which define nutrition security as “consistent access, availability, and affordability of foods and beverages that promote well-being and prevent disease.”.
Many factors impinge on the representativeness of the findings: small sample size, summer camp context, a majority of Hispanic participants, but also only one 24h food recall, which may be sufficient for large population groups but not for small groups or individuals, which would require three recalls.
We agree that the representativeness of our findings is limited by multiple design factors, including the single-site summer program setting, the predominance of Hispanic participants, and the modest sample size. We have revised the Limitations section, aligning with this comment and another reviewer's to read as follows:
As such, findings should be interpreted with caution, as they may reflect limited statistical power rather than a true absence of effect. To aid interpretation despite these limitations, Bland-Altman plots were included to provide a visual representation of agreement and variability between dietary measures. In addition, only a single 24-hour dietary recall was collected per participant, which may be sufficient for estimating average intake in large populations but is suboptimal for assessing individual dietary intake or for analyses involving small subgroups. This limitation may introduce day-to-day variability that is not accounted for in our comparison.
The short method appears valid only for girls until further validation work is done on a larger and more representative sample, using as reference 3 24h recalls instead of one. So what is the practical relevance and importance of this paper?
While we acknowledge that the SPAN HEI showed stronger performance among girls, this study offers a necessary first step in examining its potential as a feasible, low-burden dietary assessment tool in youth populations. We have added the following in the discussion section to highlight this:
Although the strongest alignment between the SPAN HEI and HEI-2020 was observed among girls, the overall findings provide important preliminary evidence supporting the potential utility of the SPAN HEI for dietary surveillance in youth. Given the increasing demand for low-burden, scalable tools to monitor dietary behaviors, particularly in school and community settings, this work contributes to the growing evidence base. Further validation in more diverse and representative samples, using multiple 24-hour recalls, is warranted.
Reviewer 3 Report
Comments and Suggestions for Authors
The authors present an interesting study in which two approaches to dietary assessment were examined with respect to one another to determine the efficacy of both approaches. Briefly, the authors deployed both assessments within a particular grade cohort, with the results of each crosschecked to identify trends/differences/outliers. Overall, statistics show that there was a degree of correlation between both assessments despite different approaches, with the data suggesting that the more streamlined of the two may be a more beneficial means of measuring dietary trends at the population level given the more simplistic measurement style, though further research is required.
In reviewing the manuscript I made a couple of observations. the following should be considered by the authors when preparing a suitable revision.
- The study aimed to eliminate variables such as the day of the week the assessments took place, the proximity to holidays, etc. but I would wonder did the authors consider the time of the year and whether this mattered in the context of the study? Dietary habits can vary throughout the year relative to season, trends, etc. and I would wonder whether this was anticipated by the authors at all and whether they can comment on such.
- The methods section would benefit from being ‘broken up’ under headings for each phase of the study.
- Moreover, it might be clearer if a graphic of sorts demonstrating the workflow was produced to show the process overall. It would also make it clearer if the demographics were included as part of this in terms of data included/data excluded at each phase.
- Can the authors comment on how 12 classrooms ultimately only produced 51 responses? This seems a low number all things considered. Were there particular factors that prevented participation? Could this be attributable to one assessment over another?
Author Response
We thank the editor and the peer reviewers for their time and expertise in the review of our submission. We have reviewed and responded to each comment provided by both peer reviewers below. Responses to comments are provided and bolded. The resubmitted documents and manuscripts (both clean and with track changes) provides all additional information and addressed feedback from reviewers using track changes.
Reviewers' comments to author:
Reviewer 3.
The authors present an interesting study in which two approaches to dietary assessment were examined with respect to one another to determine the efficacy of both approaches. Briefly, the authors deployed both assessments within a particular grade cohort, with the results of each crosschecked to identify trends/differences/outliers. Overall, statistics show that there was a degree of correlation between both assessments despite different approaches, with the data suggesting that the more streamlined of the two may be a more beneficial means of measuring dietary trends at the population level given the more simplistic measurement style, though further research is required.
In reviewing the manuscript I made a couple of observations. the following should be considered by the authors when preparing a suitable revision.
The study aimed to eliminate variables such as the day of the week the assessments took place, the proximity to holidays, etc. but I would wonder did the authors consider the time of the year and whether this mattered in the context of the study? Dietary habits can vary throughout the year relative to season, trends, etc. and I would wonder whether this was anticipated by the authors at all and whether they can comment on such.
We appreciate the reviewer’s attention to seasonal variability in dietary habits. This study used a convenience sample recruited during a structured summer program, which closely resembled the daily routine and environment of a typical school year (e.g., scheduled meals, supervised activities). While this setting helped control for some variability, we acknowledge that seasonality and unmeasured contextual factors (e.g., holiday proximity, weather patterns) could influence dietary intake. We have added language to the limitations section noting that these factors could not be fully accounted for in this small pilot study.
The methods section would benefit from being ‘broken up’ under headings for each phase of the study.
We agree that the Methods section is extensive; however, it was structured in accordance with the MDPI Nutrients journal guidelines, which recommend a unified “Materials and Methods” section without numbered subheadings.
Moreover, it might be clearer if a graphic of sorts demonstrating the workflow was produced to show the process overall. It would also make it clearer if the demographics were included as part of this in terms of data included/data excluded at each phase.
We appreciate the reviewer’s suggestion regarding the addition of a workflow graphic. It is our hope that all key methodological steps and inclusion criteria are now clearly outlined in the revised Methods section. We believe the updated narrative provides sufficient transparency, but we remain open to further editorial direction if a visual is ultimately required.
Can the authors comment on how 12 classrooms ultimately only produced 51 responses? This seems a low number all things considered. Were there particular factors that prevented participation? Could this be attributable to one assessment over another?
The study was conducted within a small summer program and relied on a convenience sampling approach. While 12 classrooms were involved, final participation was limited by voluntary enrollment, required parent/guardian consent, student assent, and some absences during the assessment period. We also acknowledge that the dual-assessment structure may have contributed to reduced participation. These issues are now addressed in the revised Limitations section, where we also emphasize the need for future studies with larger, representative samples and adequate power.
Round 2
Reviewer 2 Report
Comments and Suggestions for Authors
I insist that the authors use food security instrad of nutrition security, referring to appropriate references by WHO/FAO. Food security encompasses access to a nutritionally adequate diet but nutrition security also implies that food nutrients can be adequately handled by the body because of health.
Author Response
We thank the editor and the peer reviewers for their time and expertise in the review of our submission. We have reviewed and responded to each comment provided by both peer reviewers below. Responses to comments are provided and bolded. The resubmitted documents and manuscripts (both clean and with track changes) provides all additional information and addressed feedback from reviewers using track changes.
Reviewers' comments to author:
Reviewer 2.
I insist that the authors use food security instead of nutrition security, referring to appropriate references by WHO/FAO. Food security encompasses access to a nutritionally adequate diet but nutrition security also implies that food nutrients can be adequately handled by the body because of health.
We thank the reviewer for their thoughtful follow-up. We have edited the manuscript to incorporate food security as it is referenced by WHO/FAO.
Reviewer 3 Report
Comments and Suggestions for Authors
The authors have given more or less adequate rebuttal to the points raised. While I do appreciate there are certain limitations from an experimental design perspective, but also the journal in terms of formatting, I do believe that there are small changes that could be implemented to improve on these points. For example, the participation rate is low, and yes, there are obstacles such as volunteering, getting consent etc. - this should possibly be highlighted as a limitation more strongly so that it is clearer to any subsequent reader who may find this study.
Author Response
We thank the editor and the peer reviewers for their time and expertise in the review of our submission. We have reviewed and responded to each comment provided by both peer reviewers below. Responses to comments are provided and bolded. The resubmitted documents and manuscripts (both clean and with track changes) provides all additional information and addressed feedback from reviewers using track changes.
Reviewers' comments to author:
Reviewer 2.
Reviewer 3.
The authors have given more or less adequate rebuttal to the points raised. While I do appreciate there are certain limitations from an experimental design perspective, but also the journal in terms of formatting, I do believe that there are small changes that could be implemented to improve on these points. For example, the participation rate is low, and yes, there are obstacles such as volunteering, getting consent etc. - this should possibly be highlighted as a limitation more strongly so that it is clearer to any subsequent reader who may find this study.
We thank the reviewer for this helpful suggestion. We agree that the low participation rate is an important limitation that should be clearly stated for future readers. We have strengthened the Limitations section to more explicitly address this issue, noting the barriers posed by the voluntary nature of participation and consent requirements that go beyond the statistical limitations. We also clarify how these factors may affect generalizability.
“Participation in the study was voluntary and required both parent/guardian consent and student assent, which presented logistical barriers that likely contributed to the low enrollment rate. This limited participation should be interpreted as a constraint on the generalizability and scale of the findings and underscores the challenge of recruiting youth in school or community-based settings.”